# Hesperidin Reversed Long-Term *N*-methyl-*N*-nitro-*N*-Nitroguanidine Exposure Induced EMT and Cell Proliferation by Activating Autophagy in Gastric Tissues of Rats

**DOI:** 10.3390/nu14245281

**Published:** 2022-12-11

**Authors:** Zhaofeng Liang, Jiajia Song, Yumeng Xu, Xinyi Zhang, Yue Zhang, Hui Qian

**Affiliations:** 1Wujin Institute of Molecular Diagnostics and Precision Cancer Medicine of Jiangsu University, Wujin Hospital Affiliated of Jiangsu University, Changzhou 213017, China; 2Department of Laboratory Medicine, School of Medicine, Jiangsu University, Zhenjiang 212013, China

**Keywords:** gastric cancer, MNNG, hesperidin, autophagy, PI3K/AKT

## Abstract

Gastric cancer is a common malignant tumor worldwide. *N*-methyl-*N*-nitro-*N*-nitroguanidine (MNNG) is one of the most important inducing factors of gastric cancer. Autophagy can affect the occurrence and development of gastric cancer, but the mechanism is not clear. Chemoprevention has been shown to be a rational and very promising approach to the prevention of gastric cancer. Hesperidin is a citrus flavone, an abundant polyphenol in citrus fruits and traditional Chinese medicine. It has an excellent phytochemistry that plays an intervention role in gastric cancer. However, it is unclear whether long-term exposure to MNNG will affect the occurrence of gastric cancer by regulating autophagy and whether hesperidin can play an intervention role in this process. In the present study, we demonstrated that long-term MNNG exposure inhibits autophagy in stomach tissues of rats, promotes the epithelial–mesenchymal transition (EMT) process and cell proliferation and suppresses the activity of the PI3K/AKT pathway. We further found that after rapamycin-activated autophagy, long-term MNNG exposure promoted cell proliferation and EMT were inhibited. In addition, hesperidin promotes autophagy and the activity of the PI3K/AKT pathway, as well as the suppression of proliferation and EMT in the stomach tissues of rats. Our findings indicate that hesperidin reverses MNNG-induced gastric cancer by activating autophagy and the PI3K/AKT pathway, which may provide a new basis for the early prevention and treatment of MNNG-induced gastric cancer.

## 1. Introduction

Gastric cancer is the fourth most common malignant cancer and the third most common cause of cancer death worldwide [1]. Many risk factors are related to the occurrence and development of gastric cancer, including dietary factors, infection, environmental factors, and gene mutation [2,3]. N-nitrosamines are one of the most important inducing factors of gastric cancer, which widely exist in the environment and food. N-methyl-N-nitro-N-nitroguanidine (MNNG) is a common N-nitrosamine, which is often used to simulate the exposure of N-nitrosamine in the living environment and common food. It has been confirmed that MNNG induces precancerous lesions of gastric cancer in vivo and in vitro models [4,5]. Studies also show that MNNG promotes the occurrence of gastric cancer [6,7]. Although great progress has been made in understanding the relationship between MNNG and gastric cancer, the underlying mechanisms are still unclear. Abnormal cell proliferation and EMT play a critical role in gastric cancer and can be induced by various harmful stimuli [8,9,10,11].

EMT makes cells lose epithelial characteristics and obtain mesenchymal characteristics, which contribute to the progression of cancer. It has been reported that cells or mice exposed to MNNG promote the EMT process. Cancer is also characterized by abnormal cell proliferation [12,13]. Studies indicate that exposure to MNNG induces cell proliferation [14,15]. Nevertheless, the molecular mechanism of MNNG-induced gastric EMT and cell proliferation is unclear. Autophagy plays an important role in the occurrence and development of gastric cancer, but it is unclear whether MNNG regulates autophagy to participate in the occurrence of gastric cancer.

Autophagy is a conserved self-defense mechanism that responds to various cellular stresses [16,17,18]. Normal and moderate autophagy is particularly important for cell survival and maintaining homeostasis. Dysregulation of autophagy involves the occurrence and development of a variety of diseases, including gastric cancer. Increasing pieces of evidence show that autophagy plays a critical role in the occurrence and development of gastric cancer [17,19,20]. Studies demonstrated that autophagy plays a regulatory role in the occurrence and development of cancer by regulating EMT and abnormal cell proliferation [21,22,23]. Autophagy is regulated by different signaling pathways (PI3K/AKT, P53, MAPK, PTEN, AMPK, etc.) and ncRNAs (miRNA, lncRNA, circRNAs)[16,17,18,24,25]. Therefore, investigating the mechanism of MNNG-mediated gastric autophagy, EMT, and abnormal cell proliferation may provide new insights for the early treatment and intervention of gastric cancer.

The incidence of gastric cancer is hidden, and there are no characteristic early diagnostic markers in the clinical setting. It is difficult to diagnose gastric cancer early, so the intervention of phytochemicals is particularly important. Phytochemicals have excellent anti-cancer activity and health benefits. Hesperidin is a multi-effect citrus flavone, and the most abundant polyphenol in citrus fruits and traditional Chinese medicine [26,27]. Studies have confirmed that hesperidin has minimal or no side effects [28] and has an anti-cancer effect in a variety of cancers, such as gastric cancer [29,30], breast cancer [27], and lung cancer [31,32]. Although the role of hesperidin’s anti-cancer activity in different cancer has been studied, limited work has been conducted on its role in MNNG-induced gastric cancer. Choi et al. reported that hesperidin has a protective effect on the TGF-β1 elicited EMT-like changes of podocytes through regulation of the mTOR pathway [33]. Wang et al. found that hesperidin suppresses EMT-mediated invasion and migration of cervical cancer cells by inhibiting abnormal activation of TGF-β1/Smads pathway [34]. It was reported that hesperidin could inhibit the cell proliferation of gastric cancer cells by activating the mitochondrial pathway [35]. Jeong et al. found that hesperidin inhibits the cell proliferation of prostate cancer cells via inducing oxidative stress and destroying Ca2+ homeostasis [36]. Studies have shown that hesperidin can improve mitochondrial damage and promote gastric motility in functional dyspepsia rats by regulating the phagocytosis of interstitial cells of cajal mitochondria mediated by Drp1[37]. The results of Li et al., demonstrated that hesperidin reduced myocardial ischemia/reperfusion injury by suppressing excessive autophagy [38]. However, it is still unclear whether hesperidin can play an intervention role in gastric cancer induced by long-term MNNG exposure by regulating cell autophagy, cell proliferation, and the EMT process.

Herein, in order to explore whether hesperidin can play an intervention effect and its molecular mechanism in MNNG-induced gastric cancer through regulating cell autophagy, cell proliferation, and EMT, we designed this study. In the current study, we established a rat model of long-term MNNG exposure, observed the effects of long-term MNNG exposure on autophagy, EMT, and cell proliferation and observed whether the PI3K/AKT pathway is involved in this process. The preventive effects of hesperidin are also determined in the gastric tissues of long-term MNNG-exposed rats. The findings of the current study may provide a way forward in the pathogenesis and early intervention of MNNG-induced gastric tumorigenesis.

## 2. Materials and Methods

### 2.1. Chemicals and Reagents

E-cadherin antibodies were purchased from Cell Signaling Technology (Danvers, MA, USA). Beclin1, LC3-I/II, ATG5, *p*-PI3K, *p*-Akt, vimentin, and proliferating cell nuclear antigen (PCNA) were purchased from Bioworld (Minnesota, Billerica, MA, USA). GAPDH was from Biogot Technology (Nanjing, China), Horseradish peroxidase-conjugated secondary antibodies were purchased from Bioworld (Minnesota, USA), and the primers for vimentin, E-cadherin, and PCNA were synthesized according to published sequences and synthesized by Invitrogen (Carlsbad, CA, USA). MNNG (purity ≥ 98%) was purchased from Chroma Biotechnology Co. Ltd. (Chengdu, China). Rapamycin was purchased from Med Chem Express (New Jersey, NJ, USA). Sources of other materials are noted accordingly in the text.

### 2.2. Rats and Exposure to MNNG

Male Wistar rats (*n* = 60), 6–8 weeks old, were purchased from the Animal Research Center of Jiangsu University. Rats were housed in polypropylene cages at 20–22 °C and 40–60% humidity with 12 h light/dark cycles. All of the mice experiments were approved by the Animal Care and Use Committee of Jiangsu University, and efforts were made to minimize suffering and distress. Wistar rats were acclimated for 1 week prior to MNNG, rapamycin, and hesperidin exposure. The control group rats had free access to water and a normal diet (Open Formula Animal Diets, Changzhou, China). The rats in the MNNG group received MMNG by intragastric administration (100 mg/kg BW) every 5 days for 12 weeks [6]. After the final MNNG exposure, rats were sacrificed, and gastric tissues were collected and stored at −80 °C for further analysis.

### 2.3. In Vivo Delivery of Rapamycin

Rapamycin is a commonly used autophagy activator. To study the role of autophagy in cell proliferation and EMT induced by long-term MNNG exposure, we treated rats with rapamycin. Wistar rats 6–8 weeks old were grouped according to the following: the control group rats had free access to water and a normal diet; the MNNG group rats were exposed to MNNG; the MNNG and DMSO group, rats were injected with sterile DMSO (Sigma-Aldrich, St. Louis, MO, USA) and exposed to MNNG; the MNNG and rapamycin group rats were injected with rapamycin (1 mg/kg BW) and exposed to MNNG. Rapamycin was administered intraperitoneally three times a week by dissolving it in the sterile DMSO. Rats were weighed weekly. After the last treatment, rats were sacrificed, and gastric tissues were collected and stored at −80 °C for further analysis.

### 2.4. Hesperidin Intervention

In order to explore whether hesperidin can play an intervention role and s in MNNG-induced gastric cancer, we treated rats with hesperidin (30 mg/kg BW). Wistar rats 6–8 weeks old were grouped into 4 groups: the control group rats received a normal diet; the MNNG group rats were exposed to MNNG and received a normal diet; the MNNG and solvent control group rats were exposed to MNNG and DMSO and received a normal diet; the MMNG and hesperidin group rats were exposed to MNNG and received 30 mg/kg hesperidin by gavage and a normal diet. After the 12 weeks treatment, rats were sacrificed, and gastric tissues were collected and stored at −80 °C for further analysis.

### 2.5. Hematoxylin–Eosin Staining

Stomach tissues of rats in the control group and different treatment groups were fixed in 4% buffered formalin for 24 h, then paraffin-embedded for slicing into 5 μm sections. The sections were used for subsequent hematoxylin–eosin staining. Subsequently, sections were deparaffinized with xylene and rehydrated in water through a series of different grades of alcohol. The gastric tissues of rat sections were stained with hematoxylin–eosin staining.

### 2.6. Immunohistochemical Staining

Immunohistochemistry was performed according to the standard protocol. Briefly, 5 μm paraffin-embedded stomach tissues sections of rats in the control group and different treatment groups were de-waxed in xylene and rehydrated in an alcohol gradient. Endogenous peroxidase activity was inhibited with exposure to 3% (*v*/*v*) H_2_O_2_ solution. Sections were boiled in citrate buffer (pH 6.0, 10 mM) for antigen retrieval and incubated with 5% (*w*/*v*) BSA to block the non-specific binding. The sections were incubated with the primary antibody (E-cadherin and vimentin) at 4 °C overnight and subsequently incubated with a biotinylated secondary antibody. Finally, sections were incubated with biotinylated immunoglobulin G and SABC (BOSTER, China). Images were taken using a Nikon Eclipse Ti-S microscope (Nikon Corporation, Tokyo, Japan) at X200 magnification.

### 2.7. RNA Extraction and Real-Time PCR

Total RNA was isolated from gastric tissues of rats in the control group and different treatment groups using Trizol (Gibco, CA, USA). Reverse transcription was performed according to the manufacturer’s protocol of the reverse transcription kit (Vazyme, Nanjing, China); 1 μg RNA was used for reverse transcription. Real-time PCR was carried out on the step one plus real-time PCR System (ABI, USA) by using the AceQ QPCR SYBR GREEN master mix (Vazyme, China). GAPDH served as the housekeeping gene. Fold changes of gene expression were evaluated by the 2−ΔΔCt method. The primers were synthesized by biological companies (Invitrogen, MA, USA), and the sequences of the primers are listed in Table 1.

### 2.8. Western Blotting

Gastric tissues of rats in the control group and different treatment groups were homogenized using a fully automatic sample rapid grinding instrument (Shanghai Jingxin Industrial Development, Co., Ltd., Shanghai, China) in lysate buffer 1% protease inhibitors (Roche, Basel, Switzerland) and centrifuged at 4 °C for 20 min. Equal amounts (60 μg) of the protein were separated on 7.5–10% SDS-polyacrylamide gel and transferred to PVDF membranes (Millipore, Burlington, MA, USA). After being blocked with skim milk (5%) at 25 ˚C for 1 h, the PVDF membranes were incubated with the primary antibodies (Beclin1, LC3-I/II, ATG5, *p*-PI3K, *p*-Akt, vimentin, and PCNA) at 4 °C overnight. After being washed with tris-buffered saline and Tween 20, the PVDF membranes were incubated with horseradish peroxidase-conjugated secondary antibody, and bands were detected using an enhanced chemiluminescence kit (Millipore, USA).

### 2.9. Statistical Analysis

All the statistical data were presented as mean ± standard deviation and analyzed by using SPSS software 22.0 (SPSS, Chicago, IL, USA). ANOVA was used for the comparison of statistical differences among multiple groups, followed by the multiple comparisons (LSD) significant difference tests. The Unpaired Student’s test was also used for the comparison between two groups. Values of *p* < 0.05 were considered significant.

## 3. Results

### 3.1. Long-Term MNNGExposure Induced EMT and Cell Proliferation and Inhibited Autophagy in the Gastric Tissues of Rats

N-nitrosamine exposure is one of the most important risk factors for gastric cancer. Abnormal EMT and cell proliferation are crucially involved in the initiation of gastric cancer. Autophagy plays a regulatory role in the occurrence and development of cancer by regulating EMT and abnormal cell proliferation. After 12 weeks of MNNG exposure, we assessed the changes in histology alterations and the changes in the expression levels of autophagy, EMT, and cell proliferation markers in the gastric tissues of rats. Histological analysis showed that long-term MNNG exposure induced a canceration tendency in gastric tissue of rats (Figure 1A). The results of real-time PCR showed that long-term MNNG exposure reduced E-cadherin and Beclin1 mRNA levels, and the expression of vimentin and PCNA were elevated (Figure 1B). Western blot analyses revealed that long-term MNNG exposure reduced Beclin1, LC3-I/II, ATG5, and E-cadherin expression, while the levels of vimentin and PCNA increased (Figure 1C). Immunohistochemical staining also showed that long-term MNNG exposure increased vimentin expression and decreased Beclin1 levels (Figure 1D). These results suggested that long-term MNNG exposure induces autophagy, EMT, and proliferation in the gastric tissues of rats. Long-term MNNG exposure may play an important role in the occurrence and development of gastric cancer by affecting autophagy, cell proliferation, and the EMT process.

### 3.2. Long-Term MNNG Exposure Decreased the PI3K/AKT Pathway Activation in Rats Gastric Tissues

In order to clarify the molecular mechanism of autophagy, EMT, and cell proliferation of gastric tissue cells induced by long-term MNNG exposure, we analyzed whether gastric autophagy, EMT, and cell proliferation triggered by long-term MNNG exposure correlate with the PI3K/AKT pathway in the gastric tissues of rats; the levels of phosphorylated PI3K and phosphorylated AKT were measured. The results of Western blot showed that long-term MNNG exposure decreased the expression of phosphorylated PI3K and AKT (Figure 2), suggesting that long-term MNNG exposure induces autophagy, EMT, and proliferation in the gastric tissues of rats and relates with the PI3K/AKT pathway.

### 3.3. Long-Term MNNG Exposure Mediated EMT and Proliferation Reversed by Autophagic Activation

Studies have shown that changes in autophagic activity can affect cell proliferation and the EMT process. To verify the effect of autophagy in long-term MNNG exposure induced gastric EMT and cell proliferation in the gastric tissues of rats, the rats were treated with the autophagy activator rapamycin (1 mg/kg BW, three times a week). After 12 weeks treatment with MNNG and rapamycin, Western blot results showed that rapamycin significantly increased the autophagy-related proteins Beclin1, LC3-I/II, and ATG5 (Figure 3). The above results revealed that autophagy inhibited by long-term MNNG exposure was partially recovered by rapamycin in the gastric tissues of rats.

The expression of EMT and cell proliferation markers were also detected after 12 weeks of treatment. The results of Western blot showed that the reduction of E-cadherin protein level and increase of vimentin and PCNA protein levels induced by MNNG exposure were reversed by autophagic activation (Figure 4A). Real-time PCR analyses revealed that MNNG reduced E-cadherin expression and increased the levels of vimentin, and PCNA was also reversed by autophagic activation (Figure 4B). These data indicated that activation of the autophagic pathway reverses long-term MNNG-exposure-mediated EMT and cell proliferation in the gastric tissues of rats.

### 3.4. Hesperidin Activated Autophagy and Reversed EMT, Proliferation in Rats Gastric Tissues Elicited by Long-Term MNNG Exposure

In recent years, the intervention effect of phytochemicals in gastric cancer and other cancers has received increasing attention. Hesperidin has an excellent phytochemistry that plays an intervention role in gastric cancer and other cancers. To explore the effects of hesperidin in long-term MNNG-exposure-mediated autophagy, EMT, and cell proliferation in the gastric tissues of rats, the rats were given hesperidin (30 mg/kg BW) and exposed to MNNG for 12 weeks. Results in Figure 5 indicate that hesperidin reduced expression of Beclin1, LC3-I/II, and ATG5 and reversed the effects of long-term MNNG exposure. In addition, the reduction of E-cadherin and the upregulation of vimentin and PCNA triggered by long-term MNNG exposure were also attenuated by hesperidin (30 mg/kg BW) (Figure 6). These results indicate the protective effects of hesperidin on long-term MNNG-exposure-mediated autophagy, EMT, and cell proliferation in gastric tissues of rats.

### 3.5. Hesperidin Attenuated Long-Term MNNG Exposure Activated PI3K/AKT Pathway

In this study, we found that PI3K/AKT pathway was involved in the process of long-term MNNG-induced autophagy, cell proliferation, and EMT in rat gastric tissues. To investigate whether hesperidin reverses long-term MNNG-exposure-induced alterations of gastric autophagy, EMT, and proliferation through the PI3K/AKT pathway, we further observed the alterations of phosphorylated PI3K and AKT in the gastric tissues of rats after hesperidin and MNNG exposure. We found that hesperidin (30 mg/kg BW) reversed the changes of phosphorylated PI3K and AKT induced by long-term MNNG exposure (Figure 7).

## 4. Discussion

Long-term N-nitrosamine exposure is the primary cause of gastric cancer, which promotes the initiation and progression of gastric tumorigenesis. However, the underlying mechanisms by which N-nitrosamines cause gastric cancer remain to be well-established. In the present study, we revealed that MNNG, a common N-nitrosamine chemical, induced autophagy, EMT, and cell proliferation in the gastric tissues of Wistar rats. Most importantly, we demonstrated that autophagy prevents long-term MNNG-exposure-triggered EMT and cell proliferation and promotes the PI3K/AKT pathway participation in this process. In addition, our data indicated that hesperidin promotes the PI3K/AKT pathway to prevent long-term MNNG-exposure-induced changes, such as inhibited autophagy, induced EMT process, and increased cell proliferation. These findings suggest an important role of autophagy and the intervention effect of hesperidin in long-term MNNG-exposure-mediated gastric EMT process and cell proliferation, which provide critical information for the mechanisms of long-term MNNG-exposure-related gastric tumorigenesis, as well as for the search for the potential target of gastric cancer intervention.

The EMT process, uncontrolled autophagy, and abnormal cell proliferation can directly contribute to malignant phenotypes of cancer cells. A number of studies found that uncontrolled autophagy, cell proliferation, and EMT play a critical role in the initiation and progression of gastric cancer and can be induced by various harmful stimuli [8,9,10,11]. To determine whether reduced autophagy, cell proliferation, and EMT occurred in the gastric tissues of rats exposed to MNNG for 12 weeks, the expression of autophagy, EMT, and cell proliferation markers were examined. Results of Western blot and real-time PCR showed that long-term MNNG exposure reduced the expression levels of Beclin1, LC3-I/II, ATG5, and E-cadherin and increased the levels of vimentin and PCNA. Immunohistochemical staining also showed that long-term MNNG exposure increased vimentin expression and decreased Beclin1 expression. These results suggest that long-term MNNG exposure reduces autophagy but increases EMT and cell proliferation in the gastric tissues of rats. Long-term MNNG exposure may play an important role in the occurrence and development of gastric cancer by affecting autophagy, cell proliferation, and EMT process. However, the mechanisms by which MNNG induces a reduction in autophagy, and increases cell proliferation and EMT are not well articulated.

Several signal pathways are involved in autophagy, including PI3K/AKT, P53, MAPK, PTEN, and AMPK. The PI3K/AKT pathway plays a critical role in multiple physiological processes and pathologies, including autophagy, cell proliferation, differentiation, apoptosis, inflammation, EMT, and tumorigenesis [39,40,41,42]. In our study, results showed that long-term MNNG exposure reduces autophagy but promotes EMT and cell proliferation, which is related to the downregulation of phosphorylated PI3K and phosphorylated AKT. The PI3K/AKT pathway may play an important role in the alteration of autophagy, cell proliferation, and EMT induced by long-term MNNG exposure.

The EMT and cell proliferation triggered by long-term MNNG exposure were associated with the inhibition of autophagy. To identify the role of autophagy in gastric EMT and cell proliferation, rats were treated with rapamycin (1 mg/kg BW, three times a week). Rapamycin is a commonly used autophagy activator [43,44]. After treatment with MNNG and rapamycin for 12 weeks, we found that 1 mg/kg rapamycin significantly upregulated the autophagy-related proteins Beclin1, LC3-I/II, and ATG5 in the gastric tissues of rats. These results revealed that autophagy inhibited by long-term MNNG exposure was partially recovered by rapamycin. Furthermore, the activation of autophagy reversed MNNG-mediated EMT and cell proliferation in the gastric tissues of rats, as showed by an elevated level of E-cadherin and decreased expression of vimentin and PCNA. These results imply that long-term MNNG exposure mediates EMT and cell proliferation, which is reversed by the activation of autophagy.

Phytochemicals have been attracting attention due to their potent anti-cancer activity and health benefits, especially in high-risk populations. In recent years, the intervention effect of phytochemicals in gastric cancer and other cancers has received increasing attention. Hesperidin is a multi-effect citrus flavone, and the most abundant polyphenol in citrus fruits and traditional Chinese medicine [26,27]. The safety and anticancer activity of hesperidin have been demonstrated in many studies [28,29,35]. Hesperidin has excellent antibacterial, antioxidant, and anti-cancer effects. A study by Kim HW indicated that hesperidin has antibacterial effects against *Helicobacter pylori*, thereby delaying the occurrence and development of gastric cancer [45]. It is also reported that hesperidin has a good preventive and therapeutic effect on chronic gastric ulcers induced by acetic acid in rats [46]. Other studies have also demonstrated the intervention or protective role played by hesperidin in the occurrence and development of gastric cancer [29]. However, the role of hesperidin in the development of gastric cancer induced by long-term exposure to MNNG remains unclear. Data show that hesperidin plays an intervention role by influencing autophagy, cell proliferation, and EMT process in other cancers and diseases [34,35,36,38,47,48]. However, it is still unclear whether hesperidin plays an intervention role in gastric cancer induced by long-term MNNG exposure by regulating cell autophagy, cell proliferation, and the EMT process.

The intervention effect of hesperidin on long-term MNNG-exposure-induced gastric autophagy, EMT, and cell proliferation was examined after treatment with hesperidin (30 mg/kg) and MNNG for 12 weeks. As shown in Figure 5, the reduced expression of Beclin1, LC3-I/II, and ATG5 was reversed by 30 mg/kg hesperidin in rats after long-term MNNG exposure. In addition, the reduction of E-cadherin and the upregulation of vimentin and PCNA triggered by long-term MNNG exposure were also attenuated by 30 mg/kg BW hesperidin (Figure 6). Our data showed that hesperidin exerted an intervention or protective effect in long-term MNNG-exposure-mediated autophagy, EMT, and cell proliferation in the gastric tissues of rats. This study may provide potential strategies for early intervention of gastric cancer. To further study the intervention target and molecular mechanism of hesperidin in gastric cancer induced by long-term MNNG exposure, we explored whether hesperidin exerted the intervention effect through the PI3K/AKT pathway. The results in Figure 7 demonstrate that long-term MNNG exposure triggered reduced expression of phosphorylated PI3K and phosphorylated AKT, which were attenuated by hesperidin (30 mg/kg) by increasing their expression in the gastric tissues of rats. These results of our current research provide a new potential intervention strategy and target for MNNG-induced gastric cancer.

## 5. Conclusions

Our study illustrated that autophagy inhibits long-term MNNG-exposure-triggered EMT and cell proliferation, and hesperidin exerts a protective effect on abnormal cell proliferation and EMT of gastric tissue in rats through the PI3K/AKT pathway. These findings may provide new insights into the mechanisms and the chemoprevention of MNNG-induced gastric cancer.

## Figures and Tables

**Figure 1 nutrients-14-05281-f001:**
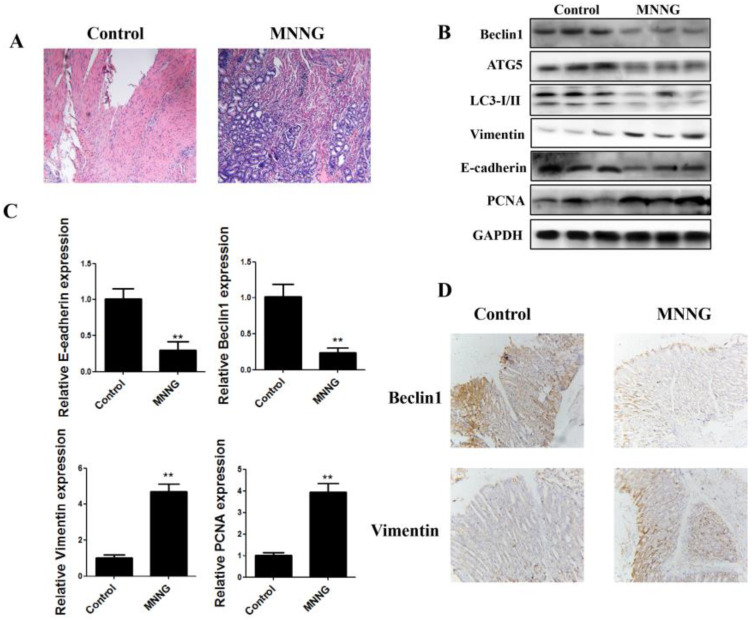
Long-term MNNG exposure induced EMT, proliferation, and inhibited autophagy in gastric tissues of rats. Rats were exposed to MNNG (100 mg/kg BW) every 5 days for 12 weeks: (**A**) hematoxylin and eosin-stained sections of gastric tissues after long-term MNNG exposure. Tissues from 3 rats were examined; the representative image is shown; (**B**) MNNG-induced alterations in the protein expression of autophagy, EMT and proliferation markers. Tissues from *n* = 3 rats as representative images; (**C**) MNNG reduced mRNA levels of autophagy and epithelial markers and increased the mesenchymal and proliferation markers. Real-time PCR results are these samples from 3 rats, and the experiments were repeated three times: (**D**) immunohistochemistry showing decreased Beclin1 expression and increased vimentin. Tissues from 3 rats were examined; the representative image is shown. GAPDH served as the housekeeping gene; ** *p* < 0.01, compared with control group.

**Figure 2 nutrients-14-05281-f002:**
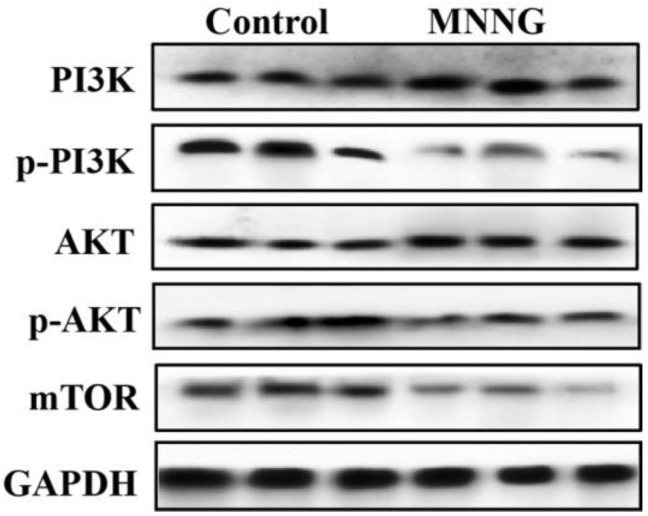
MMNG decreased the PI3K/AKT pathway activation in the gastric tissues of rats. Rats were exposed to MNNG (100 mg/kg BW) every 5 days for 12 weeks. MNNG inhibited the expressions of phosphorylated PI3K, phosphorylated AKT, and mTOR. Tissues were from *n* = 3 rats as representative images.

**Figure 3 nutrients-14-05281-f003:**
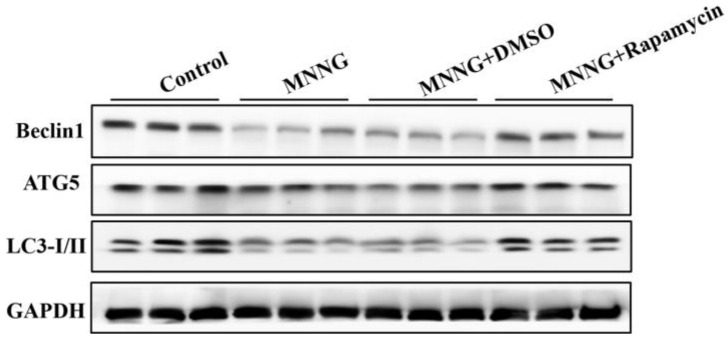
MMNG-mediated autophagy was recovered by rapamycin in the gastric tissues of rats. Rats were divided into a control group, an MNNG group, an MNNG and DMSO group, and an MNNG and rapamycin group. Western blot showed that rapamycin reversed the downregulation of Beclin1, LC3-I/II, and ATG5 induced by long-term MNNG exposure. Tissues were from *n* = 3 rats as representative images.

**Figure 4 nutrients-14-05281-f004:**
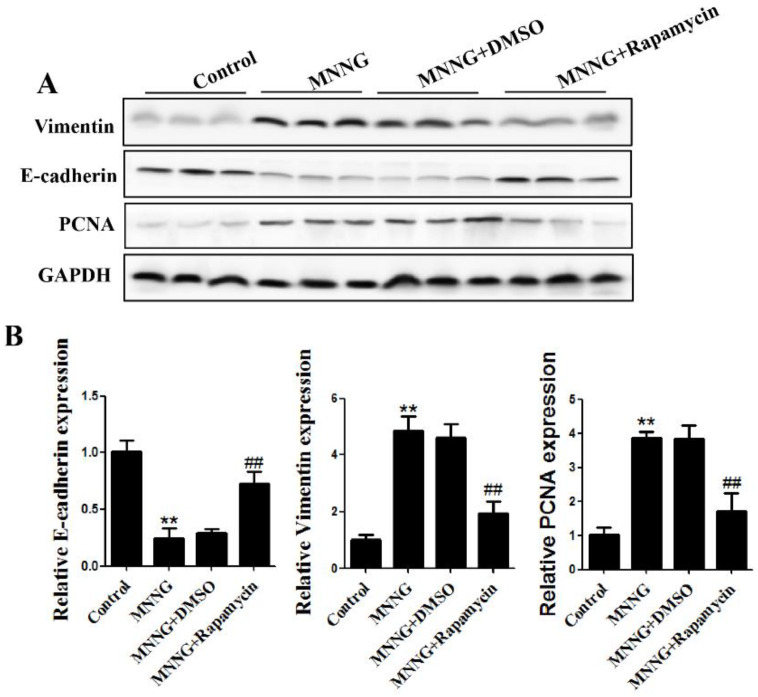
MMNG-mediated EMT and proliferation reversed by autophagy activation in the gastric tissues of rats. Rats were divided into a control group, an MNNG group, an MNNG and DMSO group, and an MNNG and rapamycin group: (**A**) Western blot analyzed the levels of EMT and proliferation markers. Tissues were from *n* = 3 rats as representative images; (**B**) real-time PCR analyzed the expression of EMT and proliferation markers. Real-time PCR results are from the samples from 3 rats and the experiment was repeated three times. GAPDH served as the housekeeping gene; ** *p* < 0.01, compared with control group; ^##^
*p* < 0.01; compared with MNNG group.

**Figure 5 nutrients-14-05281-f005:**
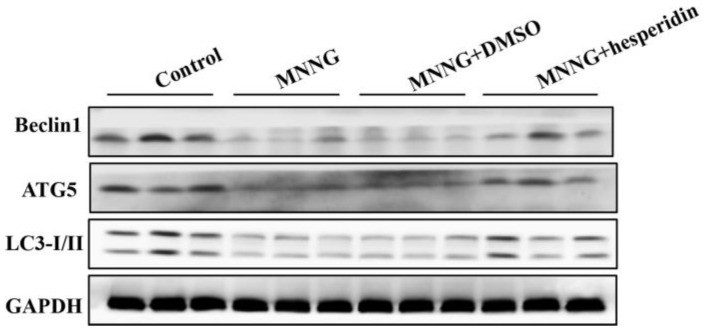
Hesperidin activated autophagy in rats’ gastric tissues elicited by MNNG. Rats were divided into a control group, an MNNG group, an MNNG and solvent control group, and an MNNG and hesperidin group. Analysis of protein expression of autophagy markers was by Western blotting. Tissues were from *n* = 3 rats as representative images.

**Figure 6 nutrients-14-05281-f006:**
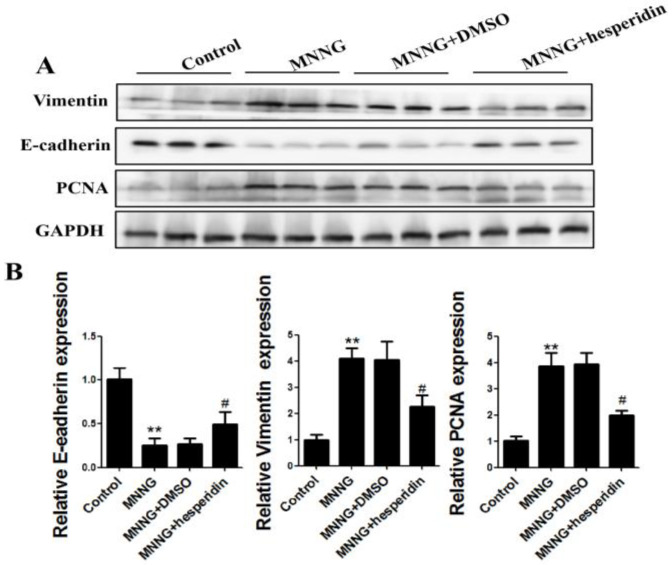
Hesperidin attenuated EMT and proliferation elicited by MNNG. Rats were divided into a control group, an MNNG group, an MNNG and solvent control group, and an MNNG and hesperidin group: (**A**) analysis of protein expression of EMT and proliferation markers by Western blotting. Tissues were from *n* = 3 rats as representative images; (**B**) real-time PCR analyzed mRNA levels of EMT and proliferation markers. Real-time PCR results are the samples from 3 rats, and the experiment was repeated three times. GAPDH served as the housekeeping gene; ** *p* < 0.01, compared with control group; ^#^
*p* < 0.05; compared with MNNG group.

**Figure 7 nutrients-14-05281-f007:**
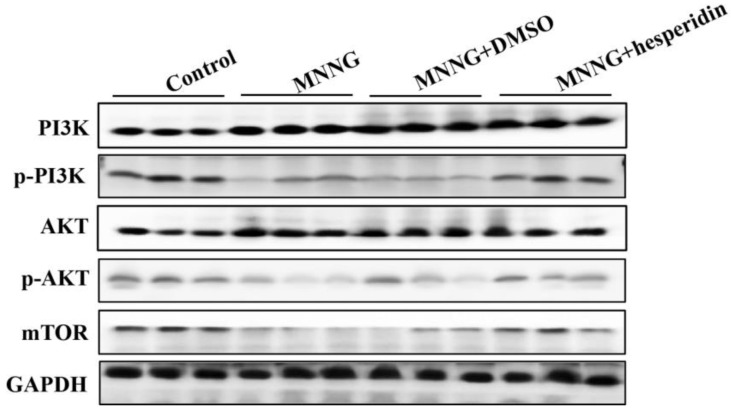
Hesperidin reversed the MNNG-inhibited PI3K/AKT pathway. Rats were divided into a control group, an MNNG group, an MNNG and solvent control group, and an MNNG and hesperidin group. Hesperidin reversed the MNNG-inhibited PI3K/AKT pathway in the gastric tissues of rats. Western blotting analyzed expression changes of phosphorylated PI3K and phosphorylated AKT. Tissues were from *n* = 3 rats as representative images.

**Table 1 nutrients-14-05281-t001:** Primer sequences.

Gene Name	Primer Sequence (5′-3′)
E-cadherin	Forward: CAGGTCTCCTCATGGCTTTGC
	Reverse: CTTCCGAAAAGAAGGCTGTCC
PCNA	Forward: CAAGAAGGTGTTGGAGGCA
	Reverse: TCGCAGCGGTAGGTGTC
Vimentin	Forward: CCTTGACATTGAGATTGCCA
	Reverse: GTATCAACCAGAGGGAGTGA
GAPDH	Forward: AGGTCGGTGTGAACGGATTTG
	Reverse: TGTAGACCATGTAGTTGAGGTCA

## Data Availability

All data generated or analyzed during this study are included in this published article.

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
