# Peer review of "Hesperidin Reversed Long-Term N-methyl-N-nitro-N-Nitroguanidine Exposure Induced EMT and Cell Proliferation by Activating Autophagy in Gastric Tissues of Rats"

_nutrients, 2022, doi:10.3390/nu14245281_

Round 1

Reviewer 1 Report

Zhaofeng Liang et al. evaluated the effect of hesperidin in the MNNG-induced gastric cancer, providing new insights into the early prevention and treatment of this cancer type. This manuscript is interesting and provides good information for readers. Nevertheless, some relevant points should be addressed.

1.     The manuscript has several grammar and stylistic issues and even typos. A deep revision is recommended since several sentences and passages are challenging to be followed.

2.     Title: I recommend revising the title carefully since its organization is confusing. In addition, the tense of the verb “reversed” can be changed to “reverses”.

3.     Abstract: It is confusing, and the order is not adequate. I suggest organizing the abstract in a structured manner.

4.     Line 84: The hesperidin origin and purity is missing and must be informed in the manuscript.

5.     The authors must follow the author’s guide on using acronyms and abbreviations.

6.     Line 124: The housekeeping gene and control tissue must be informed. control tissue) in the whole article. In addition, Table 1 and Figures 1, 3, and 6 do not mention the primers used for reference genes and the tissue selected to calculate the relative expression.

7.     Line 144: The test used to verify the normal distribution of data must be informed. What does “one-way analysis” mean? i.e., ANOVA?

8.     M&M section: This section needs to be carefully revised since some experimental details are missing. For instance, the number of replicates and the number of rat individuals are missing. In addition, the brand, model, reference, etc, of chemicals and instruments must be informed in the manuscript.

9.     Figure 3A and 3D need to be enlarged since the information is challenging to be visualized.

10.  Discussion on the hesperidin-related outcome is highly limited (lines 279-209). In the title, hesperidin effects are the center topic, but the discussion about that is only mentioned in one paragraph with 12 lines. Such a discussion must be expanded in the manuscript.

11.  Conclusions should be revised since it is very general, and some relevant, specific conclusions can be provided from the findings.

Author Response

Dear Editor:

Thank you very much for your attention to our manuscript and all the valuable comments from the reviewers and editors for our manuscript entitled “Hesperidin reversed long term MNNG exposure induced EMT and proliferation by activating autophagy and PI3K/AKT pathway in stomach tissues of rats” (nutrients-2076023). We have addressed all comments/suggestions which we believe have significantly improved the clarity and quality of our manuscript. Followings, please find our point-by-point responses to these comments.

Reviewer 1

Zhaofeng Liang et al. evaluated the effect of hesperidin in the MNNG-induced gastric cancer, providing new insights into the early prevention and treatment of this cancer type. This manuscript is interesting and provides good information for readers. Nevertheless, some relevant points should be addressed.

Response: Thank you very much for your evaluation of our manuscript.

1. The manuscript has several grammar and stylistic issues and even typos. A deep revision is recommended since several sentences and passages are challenging to be followed.

Response: Thank you for your suggestions. We have done such modifications as typos and grammar mistakes in our revised manuscript. We also invited two native English speakers to revise our manuscript.

2.Title: I recommend revising the title carefully since its organization is confusing. In addition, the tense of the verb “reversed” can be changed to “reverses”.

Response: Thank you for your comments. According to your suggestion, we have modified the title. Thanks again for your suggestions.

3. Abstract: It is confusing, and the order is not adequate. I suggest organizing the abstract in a structured manner.

Response: Thank you for your comments, which will greatly promote the quality of our manuscript. We have revised the Abstract.

4. Line 84: The hesperidin origin and purity is missing and must be informed in the manuscript.

Response: Thank you for your suggestions. We have added relevant information (the hesperidin origin and purity) to the revised manuscript.

5. The authors must follow the author’s guide on using acronyms and abbreviations.

Response: Thank you for your suggestions. We have revised the manuscript according to your suggestions.

6. Line 124: The housekeeping gene and control tissue must be informed. control tissue) in the whole article. In addition, Table 1 and Figures 1, 3, and 6 do not mention the primers used for reference genes and the tissue selected to calculate the relative expression.

Response: Thank you for your comments, which will greatly promote the quality of our manuscript. We have added relevant information in the manuscript, Table 1 and Figure Legends.

7. Line 144: The test used to verify the normal distribution of data must be informed. What does “one-way analysis” mean? i.e., ANOVA?

Response: Thank you for your comments. We have verified the statistical method and revised our description. Thanks again for your comments.

8. M&M section: This section needs to be carefully revised since some experimental details are missing. For instance, the number of replicates and the number of rat individuals are missing. In addition, the brand, model, reference, etc, of chemicals and instruments must be informed in the manuscript.

Response: Thank you for your comments, which will greatly promote the quality of our manuscript. We have added relevant information to the revised manuscript.

9. Figure 3A and 3D need to be enlarged since the information is challenging to be visualized.

Response: Thank you for your comments. But Figure 3 only shows the results of Western blot, not Fig3A and Fig3D.

10. Discussion on the hesperidin-related outcome is highly limited (lines 279-209). In the title, hesperidin effects are the center topic, but the discussion about that is only mentioned in one paragraph with 12 lines. Such a discussion must be expanded in the manuscript.

Response: Thank you for your suggestions. We have revised the manuscript according to your suggestions, especially the discussion about hesperidin. Thank you again for your comments.

11. Conclusions should be revised since it is very general, and some relevant, specific conclusions can be provided from the findings.

Response: Thank you for your comments. We have revised the conclusions according to your suggestions.

Reviewer 2 Report

Summary of the key contribution of the paper:

The Hesperidin Reversed Long Term MNNG Exposure Induced EMT and Proliferation by Activating Autophagy and PI3K/AKT Pathway in Stomach Tissues of Rats the present paper provide evidence to support the effectiveness of the strategy of discovering provide new basis for the early prevention and treatment of MNNG-induced gastric cancer.

Highlights:

·        This article clearly articulates the use of a discussion of the results of their study illustrated that autophagy regulated long term MNNG ex-posure triggered EMT and proliferation and intervention effects of hesperidin in the gastric tissues of rats.

·        The figures and tables are well referenced and clear.

·        Gastric cancer Activities application and research is up to date.

·        These findings may provide new insights into the mechanisms and the chemoprevention of MNNG induced gastric cancer.

Lowlights:

·        There are no Lowlights in this paper.

Author Response

Dear Editor:

Thank you very much for your attention to our manuscript and all the valuable comments from the reviewers and editors for our manuscript entitled “Hesperidin reversed long term MNNG exposure induced EMT and proliferation by activating autophagy and PI3K/AKT pathway in stomach tissues of rats” (nutrients-2076023). We have addressed all comments/suggestions which we believe have significantly improved the clarity and quality of our manuscript. Followings, please find our point-by-point responses to these comments.

Reviewer 2

The Hesperidin Reversed Long Term MNNG Exposure Induced EMT and Proliferation by Activating Autophagy and PI3K/AKT Pathway in Stomach Tissues of Rats the present paper provide evidence to support the effectiveness of the strategy of discovering provide new basis for the early prevention and treatment of MNNG-induced gastric cancer.

Highlights:

  • This article clearly articulates the use of a discussion of the results of their study illustrated that autophagy regulated long term MNNG ex-posure triggered EMT and proliferation and intervention effects of hesperidin in the gastric tissues of rats.
  • The figures and tables are well referenced and clear.
  • Gastric cancer Activities application and research is up to date.
  • These findings may provide new insights into the mechanisms and the chemoprevention of MNNG induced gastric cancer.

Lowlights:

  • There are no Lowlights in this paper.

Response: Thank you very much for your evaluation of our manuscript.
